# The Involvement of Innate and Adaptive Immunity in the Initiation and Perpetuation of Sjögren’s Syndrome

**DOI:** 10.3390/ijms22020658

**Published:** 2021-01-11

**Authors:** Clara Chivasso, Julie Sarrand, Jason Perret, Christine Delporte, Muhammad Shahnawaz Soyfoo

**Affiliations:** 1Laboratory of Pathophysiological and Nutritional Biochemistry, Université Libre de Bruxelles, 1070 Brussels, Belgium; clara.chivasso@ulb.be (C.C.); jason.perret@ulb.be (J.P.); christine.delporte@ulb.be (C.D.); 2Department of Rheumatology, Hôpital Erasme, Université Libre de Bruxelles, 1070 Brussels, Belgium; julie.sarrand@ulb.be

**Keywords:** sjogren’s syndrome, epithelial cells, innate immunity, lymphocytes, t cells, b cells

## Abstract

Sjogren’s syndrome (SS) is a chronic autoimmune disease characterized by the infiltration of exocrine glands including salivary and lachrymal glands responsible for the classical dry eyes and mouth symptoms (sicca syndrome). The spectrum of disease manifestations stretches beyond the classical sicca syndrome with systemic manifestations including arthritis, interstitial lung involvement, and neurological involvement. The pathophysiology underlying SS is not well deciphered, but several converging lines of evidence have supported the conjuncture of different factors interplaying together to foster the initiation and perpetuation of the disease. The innate and adaptive immune system play a cardinal role in this process. In this review, we discuss the inherent parts played by both the innate and adaptive immune system in the pathogenesis of SS.

## 1. Sjögren’s Syndrome

Sjögren’s syndrome (SS) is one of the most common autoimmune rheumatic diseases. SS is characterized by the immune-mediated destruction of exocrine glands, including lachrymal and salivary glands (SGs). Two types of SS have been defined: Primary SS (pSS), which occurs in the absence of other autoimmune diseases, and secondary SS (sSS), which is associated with other autoimmune disorders such as systemic lupus erythematosus (SLE), rheumatoid arthritis (RA), and scleroderma [1,2]. SS is characterized by a high sex preponderance with a ratio of nine female for one male. This sexual imbalance suggests an involvement of estrogens and androgens in the development of the pathology [3,4] that could account for an incidence increase of pSS during the post-menopausal stage, at the age of 40–60 years old [5]. In general, the diagnosis is based on the combination of several oral and ocular sicca symptoms, the presence of the autoimmune manifestations such the production of autoantibodies anti-Ro/SSA, the labial biopsy showing a focal lymphocytic infiltration (focus score ≥ 1 per 4 mm^2^) [6]. The pathophysiology of SS is very complex, multifactorial, and consecutive to several genetic, hormonal, environmental, and immunological risk factors. Due to its complexity, the clinical course of the pathology can be divided in several phases: An initiation phase consecutive to endogenous and exogenous factors, a dysregulation of salivary glands epithelial cells (SGECs), and an immune system activation and chronicity of inflammation induced by B cells hyperactivity [7]. The combination of all these events culminates in the destruction of the salivary gland architecture, and development of keratoconjunctivitis sicca and xerostomia. Each phase plays a significant role in the disease. The transition from the innate immune system to the adaptive system responses and the variety of cell types involved could explain the difficulties in developing an efficient therapeutic strategy for pSS. This review highlights the role of immune cells and the crosstalk between the innate and adaptive immunity in pSS pathogenesis.

## 2. Innate Immune Cells Involved in Sjögren’s Syndrome

A growing body of evidence indicates that innate immunity plays a crucial role in the pathogenesis of pSS, especially in the initiation and progression towards autoimmunity [8]. We will discuss the role of each cell type implicated in this process often called autoimmune epithelitis.

### 2.1. Dendritic Cells

Dendritic cells (DCs) are professional antigen presenting cells. They act as sentinels capturing and processing antigens, migrating in T cell areas to initiate immunity and differentiating in response to a variety of stimuli such as Toll-like receptor (TLR) ligands, cytokines, innate lymphocytes, and immune complexes [9]. DCs play a key role in pSS as they display an aberrant phenotype causing them to accumulate in SGs [10,11,12]. Saliva from pSS patients is characterized by an upregulation of C-C chemokine receptor type 5 (CCR5) and CCR5 ligands such as CC chemokine ligand type 3 (CCL3) and type 4 (CCL4) that play an important role for the effective migration of DCs to inflamed tissues. In addition, lower numbers of blood DCs in patients with pSS may be consecutive to the aberrant regulation of apoptosis [13].

Plasmacytoid DCs (pDCs) are a specific subset of DCs that can be activated by self-antigens through TLR-7 and TRL-9 [14,15] and to a lesser extent TLR-2, TRL-4, and TRL-9 [16], leading to the production of type I interferon (IFN). Type I IFN acts through autocrine and paracrine circuits sustaining a continuous reinforcing inflammatory loop. It also induces the production of the B cell activating factor (BAFF) by monocyte circulating cells and DCs contributing to the activation and differentiation of B cells into plasma cells secreting antibodies [16,17].

Follicular DCs (fDCs) originate from fibroblast precursor cells and play an essential part in the structure of ectopic germinal centers. FDCs promote B cells survival and proliferation in the long run by retaining on their surface immune-complexes (IC), formed by antigen-antibody-complement. Contrary to other DCs, fDCs do not display phagocytic activity and lack lysosomes and lysozyme [18,19].

### 2.2. Macrophages

Macrophages are the main tissue resident leucocytes and are characterized by pleomorphic phenotypes. According to their microenvironment, they can display pro-inflammatory or anti-inflammatory activities, immunogenic or tolerogenic activities, and tissue destructive or tissue regenerative activities [20,21].

In SGs specimens from patients with pSS, macrophages tend to appear early and their number is positively correlated with the biopsy focus score [22]. Macrophages are activated by interferon gamma (IFN-γ) and interleukin (IL)-17 secreted by type 1 T helper cells (Th1) and type 17 T helper cells (Th17), respectively [23]. Activated macrophages produce inflammatory cytokines such as IL-1, tumor necrosis factor alpha (TNFα), IL-18, and metalloproteases (MMPs) leading to epithelial cell damage [24,25]. Activated macrophages can also act as antigenic peptide presenting cells through their major histocompatibility complex class II (MHC-II) and interact with antigen-specific CD4+ T cells [23]. The latter, once activated, evolves in autoreactive clones that may perpetuate the activation of macrophages, themselves sustaining a pro-inflammatory auto-maintained loop [25].

Manoussakis et al., studied MSG biopsies from pSS patients and demonstrated increased infiltration by macrophages together with a marked expression of IL-18 by infiltrating macrophages. Moreover, IL-18 levels correlated with lymphoma risk factors such as persistent C4 hypocomplementemia and SG enlargement [26].

Beside SGs, this process may affect other epithelia such as the eye epithelium leading to the development of squamous metaplasia which represents the end stage of ocular involvement in pSS patients [27,28].

### 2.3. Mast Cells

Mast cells are immune cells mainly found in connective tissues. Their role in allergy and anaphylaxis is well established. However, a great deal of evidence underlines their possible involvement in tissue healing, angiogenesis, and autoimmune exacerbation [29].

In pSS patients, Leehan et al., confirmed that fibrosis of minor salivary glands (MSG) is a pathological feature of pSS that positively correlates witha focus score and is not age-related [30]. Another study identified that mast cells are strongly associated with the fibrosis and fatty infiltration of SGs. It is hypothesized that they promote fibrosis through interaction with local fibroblasts and through the production of enzymes cleaving and activating MMPs, which are essential mediators of tissue injury [31].

Mast cells express TLR-2 and TLR-4, as well as receptors for IL-1 including interleukin-1 receptor type 1 (IL1R1) and suppressor of tumorigenicity 2 (ST2). Mast cells activation through TLR-2 and TLR4 lead to the production of IL-1, TNF-α, IL-33, and chemokines such as C-X-C motif chemokine ligand 1 (CXCL1) and C-X-C motif chemokine ligand 2 (CXCL2), which recruit neutrophilic granulocytes and DCs. The activation of mast cells through IL1R1 and ST2 allows them to interact with T and B cells, interfering with antibody production [29]. The activation of ST2 on mast cells through IL-33 leads to the production of pro-inflammatory cytokines such as IL-1, IL-6, IL-13 [32], and induces a TH2 polarization of CD4+ T cells.

### 2.4. Salivary Gland Epithelial Cells (SGECs)

Salivary gland epithelial cells (SGECs) form the acinar secretory structure and the ductal excretory structures in SGs [33]. SGECs constitute the main target of auto-immunity in pSS, described as an autoimmune epithelitis [34]. Over recent years, it has become clear that SGECs also fulfill an important role in the initiation of autoimmunity.

In pSS patients, the loss of polarity of SGECs plays a crucial part in the onset of the local inflammatory process. Indeed, a decrease in occludin and zonula occludens 1 (ZO-1) expression and a redistribution of claudin to the basolateral plasma membrane have been observed in SGs from pSS patients. Furthermore, the exposure of isolated SGs cells from healthy controls to pro-inflammatory cytokines such as TNF-α and IFN-γ reproduced the alterations observed in pSS patients [35,36]. By altering the tight junction integrity of SGECs, the local cytokine production may therefore account for the secretory gland dysfunction observed in pSS patients, and subsequent decrease in saliva quality and quantity [35].

In genetically susceptible subjects, environmental stimuli such as viruses may trigger salivary gland epithelial cells (SGECs) through TLR activation [37,38]. More precisely, the activation of TLR-2 and TLR-4 expressed on the surface of SGECs results in the expression of mediators of immune activation (such as the intercellular adhesion molecule 1 (ICAM-1)), CD40, and major histocompatibility complex 1 (MHC-1) [39], as well as in IL-15 secretion inducing the proliferation of activated B and T cells and the generation and maintenance of natural killer (NK) cells. Beside leading to the expression of ICAM-1, CD40, and MHC-1, the activation of endosomal TLR3 leads to the secretion of BAFF that promotes the activation and maturation of B cells. TLR3 activation also contributes to SGEC anoikis, a form of apoptosis that is triggered by a loss of cell attachment to the extracellular matrix (ECM), thereby releasing exosomes and apoptotic blebs containing autoantigens such as Ro/SSA and La/SSB that drive autoimmunity in pSS by attracting both classical DC and pDC within SGs [40,41,42].

Activated SGECs produce chemokines that attract immune cells and contribute to the formation of germinal centers, including C-X-C motif chemokine ligand 9 (CXCL-9), C-X-C motif chemokine ligand 10 (CXCL-10), C-X-C motif chemokine ligand 12 (CXCL12), C-X-C motif chemokine ligand 13 (CXCL13) and C-C chemokine ligand 19 (CCL19), and C-C chemokine ligand 21 (CCL21) [43,44]. An increased epithelial production of cytokines such as IL-1, IL-6, and TNFα may also contribute to create a pro-inflammatory environment [45].

Activated SGECs develop the ability to act as non-professional antigen-presenting cells [46] by expressing co-stimulation molecules (CD80 and CD86) [47] and MHC-I (HLA-ABC) and MHC-II (HLA-DR), adhesion molecules such as ICAM1, vascular cell adhesion molecule 1 (VCAM-1) [46,48]. Thus, SGECs appear suitably equipped for the presentation of antigenic peptides and the transmission of activation signals to T cells [44,49].

### 2.5. Endothelial Cells

Endothelial cells, expressing CD31, form a one-cell thick walled layer called endothelium that upholster blood and lymphatic vessels. Beside bringing immunes cells to inflammation sites, endothelial cells take an active and regulatory role in inflammatory processes [50]. In response to IL-1 and TNFα, activated endothelium express adhesion molecules such as ICAM-1, VCAM-1, and E- and P-selectins that allow the interaction and migration of blood immune cells to inflamed tissues [51].

In pSS patients, the expression of ICAM-1 positively correlates with a focus score of salivary biopsies [52]. Both strong vascular endothelial growth factor C (VEGF-C) and vascular endothelial growth factor receptor 3 (VEGFR-3) expression were reported in MSGs from pSS patients [53]. As a result, increased and anatomically aberrant lymphatic neovascularization leads to a persistent extravasation of immune cells. In another study, defective lymphatic vessels were also characterized by the overproduction of CCL-21 that increased the infiltration of immune cells into inflamed tissues [54].

### 2.6. Mucosal-Associated Invariant T (MAIT) Cells

Mucosal-associated invariant T (MAIT) cells are innate-like T cells and can therefore be considered a bridge between innate and adaptive responses [55]. MAIT cells express an invariant T cell receptor (TCR) α-chain (Vα7.2–Jα33 in humans) and CD161 which is typically expressed by NK cells. They recognize vitamin B-related peptides through the evolutionary conserved non-polymorphic MHC-I-related molecule (MR1) [56]. In response to different stimuli, MAIT cells also have the capacity to express both CD4 and/or CD8 co-receptors. MAIT cells are characterized by a natural memory function and by their capacity to rapidly produce Th1, type 2 helper cells (Th2), and Th17 cytokines [55,57].

Very little is known about the contribution of MAIT cells in the pathogenesis of pSS. Wang et al. found that MAIT cells are decreased in peripheral blood circulation but are increased in SGs from pSS patients compared to healthy controls. From a functional point of view, MAIT cells from pSS patients were mainly CD4+ and naïve, in contrast with MAIT cells from controls that were almost exclusively CD8+. In addition, MAIT cells of pSS patients displayed lower levels of activation with a reduced expression of CD69 and CD154, and lower levels of TNFα and IFN-γ. The aberrant phenotype of MAIT cells in pSS patients may lead to the dysregulation of the local immune responses, which would trigger local damage in SGs and auto-immunity [57].

### 2.7. Natural Killer (NK) Cells

Natural Killer (NK) cells are a cytolytic component of the innate immune system. They have the ability to sense the pathological changes of self-cells and therefore take an important part in the immune surveillance of tumor cells and virus-infected cells [58]. NK cells express the NKp30 receptor that is recognized by DCs and lead to the production of Th1 cytokines such as IFN-γ and IL-12 [59].

NK cells are enriched in MSGs from pSS patients and their presence correlate with the focus score [60]. In addition, NK cells overexpress the NKp30 receptor and SGECs express B7-H6, the ligand for NKp30. Taken together, this may explain the hyperactivity of NK cells and the interrelation with SGECs and DCs that lead to a subsequent activation of innate and adaptive immunity. The expression of B7-H6 by SGECs may also be involved in the homing of NK cells in SGs [60]. Another study identified a subset of NK cells that expresses NKp44 and produces IL-22 in SGs from pSS patients. This subgroup has however been reclassified and is now part of Innate Lymphoid Cells (ILCs), which will be discussed in the Section 2.9 of this article [61].

### 2.8. Natural Killer T (NKT) Cells

NK T (NKT) cells are immune components that share features of both T cells and NK cells. They discriminate self from non-self-antigens and produce prompt immune responses against Gram-negative bacteria [62]. They are a major source of IL-4 and to a lesser extent of IFN [63]. Invariant NKT (iNKT) cells are a special subset of NKT cells that seems to play a pivotal role in the regulation of immunity. They express CD161 (typical of NK cells) and a semi-invariant T cell receptor (TCR). By linking CD1d in B cells with their invariant TCR, iNKT cells are able to suppress B cell auto-reactivity. In addition, under certain circumstances, they can express both CD8+ and CD4+, which leads to the production of Th1 and Th2 cytokines [62,64].

A decreased number of NKT cells was observed in the peripheral blood of pSS patients, which could be explained by apoptosis or homing in SGs [65]. Another study reported an increased number of iNKT in peripheral blood but a complete absence of iNKT cells together with an increased number of auto-reactive B cells in SGs from pSS patients [66]. These data were corroborated by showing that the lack of CD1d following B-cells hyperactivation lead to a greater release of autoantibodies [67]. In spite of the studies supporting the candidacy of NK cells in the SG of pSS patients, there is actually no sufficient proof bolstering their role as participating actively in the pathology of SS.

### 2.9. Innate Lymphoid Cells (ILCs)

Innate lymphoid cells (ILCs) are the innate counterparts of T helper lymphocytes [68,69]. They are mostly concentrated at epithelial barriers and rapidly release cytokines in response to environmental triggers. They can be classified into three subsets according to the expression of specific transcription factors and the production of cytokines that mirror the subsets of helper T. ILC1 express the T-Box Transcription Factor 21 (TBX21, also named T-bet), produce IFN-γ, and respond to intracellular pathogens such as viruses. ILC2 express the transcription factor GATA Binding Protein 3 (GATA-3), produce IL-4, IL-5 and IL-13, and respond to extracellular parasites and allergens. ILC3 express the transcription factor Retinoic acid-related orphan receptor gamma t (RORγt), produce IL-17A and IL-22, and react to extracellular pathogens such as bacteria and fungi [68,70,71].

ILCs have been identified in the SGs from pSS patients [61,72] and may contribute to the formation of germinal center-like structures [73]. Blokland et al. showed that the presence of ILC1 was associated with a higher disease activity score (ESSDAI). ILC1 could contribute to the pathogenesis of pSS through the massive production of IFN-γ, but the underlying mechanisms remain largely elusive [74]. In addition, an increased IFN signature and reduced frequencies of ILC2 and ILC3 was associated with a high expression of Fas cell surface death receptor (Fas also named CD95) by ILC2 and ILC3. It was hypothesized that the increased Fas expression on ILC2 and ILC3 may induce an apoptosis of these cells. These observations corroborate previous studies in mice that reported a link between type I IFN, pDC activation, and apoptosis of circulating ILC2 and ILC3 [75,76,77].

A subset of ILC3 that was originally classified as NK cells because of its expression of NKp44 was identified in SGs from pSS patients. It was found to be a major source of IL-22 together with Th17 cells. The frequency of ILC3 was positively correlated to the focus score [61]. Additional studies are needed to further evaluate the ILC3 function in SGs from pSS patients. Currently, their role in the pathogenesis of SS remains to be determined and detailed. There is not enough data purporting their role as being active players in SS pathology.

Figure 1 summarizes the action of the different players of innate immunity in the pathogenesis of pSS.

## 3. Adaptive Immunity: The Insidious Role of T Cell Subsets in SS

Although B cells are involved in the last chronic inflammation phase by producing autoantibodies, pSS is mostly dominated by T lymphocytes in the early stages of the disease. In the past, T cells have been considered the most abundant cells in the pathogenic picture of SS and the Th1 subset was identified as the major cell type infiltrating MSGs. Today, the discovery of other T cell subpopulations has unveiled new avenues revealing a further understanding of the pathogenesis of pSS. The infiltrating cells observed in pSS SGs biopsies are composed of about 45–50% of CD4+ T lymphocytes, 20% of CD8+ T lymphocytes, and 20% of B cells [78]. One of most common clinical features of pSS is the reduction of lymphocytic cells in peripheral blood. Several models have been proposed to explain this phenomenon but the most accredited model attributes this phenomenon to a selective migration of the CD4+ T-cell to the inflamed tissue. Although this model remains speculative since the mechanism of cell migration still remains unclear, CD4+ T-cells represent one of the significant protagonists of pSS inflammation.

### 3.1. CD4+ T Cells

#### 3.1.1. Th1-Th2 Cells

During the differentiation cascade of T CD4+, the proliferating helper T cells can differentiate into two subtypes based on distinct cytokine patterns: Th1 and Th2 cells.

Upon T-cell activation, IFN-γ and IL-12 induce the expression of T-bet and the signal transducer and activator of transcription (STAT)-4, which is involved in the differentiation of naïve CD4+ T cells into Th1 lymphocytes. Th1 cells predominantly produce pro-inflammatory cytokines such as IFN-γ [79] and IL-2. In contrast, IL-2 and IL-4 induce the GATA-3 transcription factor and the consequent polarization of naïve T cells into Th2. Th2 cells produce anti-inflammatory cytokines such as IL-4, IL-5, IL-9, IL-10, IL-13, and IL-25 [80]. Several studies have suggested that pSS was related to abnormal Th1 activation and SGs infiltration [81]. This evidence was supported by the presence of elevated levels of IFN-γ in serum and Th1 cells in blood. Furthermore, T cells expressing high levels of IFN-γ and STAT-4 mRNA have been found in SGs from SS patients while Th2-related marker transcripts were observed only in GC-positive patient SGs biopsies with severe B cell infiltration and organization. These data suggest that these two cell types operate in different stages of the disease. Paradoxically, a Th2 cytokine profile (IL-6 and IL-10) was observed in blood from pSS patients while the affected tissues were mainly characterized by Th1 and/or Th17 cells. This Th1/Th2 imbalance, generally observed in various chronic inflammatory disorders, is not easily understood because of a limited number of studies (Figure 2) [82].

#### 3.1.2. Th17

The inflamed SGs from pSS patients represent a perfect environment for the recruitment and polarization of Th17 cells. Although their contribution to the pathogenesis of pSS remains unclear, Th17 cells can fulfill several roles considered potentially pathogenic. In healthy conditions, Th17 cells play a fundamental role in maintaining mucosal barrier integrity by inducing the synthesis of tight junction proteins [83], proliferation of epithelial cells [84], and playing a defense role against microbes [85,86,87]. Th17 cells differentiate from naive CD4+ cells in the periphery in response to different signals and cytokines secreted by antigen-presenting cells [88]. The critical cytokine involved in Th17 differentiation is transforming the growth factor beta (TGFβ) in combination with IL-6 or IL-21 [88,89] which induce the activation of several transcription factors such as signal transducer and activator of transcription (STAT)-3 that induces the expression of RORγt, a member of the retinoic acid–related orphan nuclear hormone receptor family [88]. RORγt in synergy with RORα promotes Th17 differentiation [90]. Th17 cells produce IL-17 and other inflammatory cytokines such as TNF-α, IL-22, and IL-26. IL-17 also called IL-17A is a member of six cytokines family which also includes IL-17F, another Th17 specific cytokine. IL-17A and IL-17F are characterized by a 55% amino acid sequence identity and a common receptor [91]. These cytokines along with TNF-α are especially involved in inducing and mediating pro-inflammatory responses. Regarding IL-22 and IL-26, much less is known. Several studies carried out in different cohorts of sicca-SS and sicca-non SS have showed the presence of an elevated level of IL-17 cytokines and IL-17 mRNA in tears [92,93] and SGs from sicca-SS patients in association with lymphocytic infiltrates [94,95].

#### 3.1.3. T Follicular Helper Cells

T follicular helper (Tfh) cells are a subset of CD4+ T that act in collaboration with B cells to promote and regulate humoral responses. Tfh cell differentiation process requires initial interaction with DCs [96] and B cells. Tfh are mainly situated in secondary lymphoid organs and express a specific combination of cell surface molecules [97]. The best marker that defines the Tfh cells is the C-X-C motif chemokine receptor type 5 (CXCR5) [98,99] whose expression is regulated by a B-cell lymphoma 6 (Bcl-6) transcription factor. In the first stage of development, the naive CD4+ T cells do not express CXCR5 but only the C-C chemokine receptor type 7 (CCR7) that allow them to enter secondary lymphoid organs and reach the T-cell zones. Following their activation, the Tfh cells downregulate CCR7 and upregulate CXCR5, which facilitates their migration from the T-cell zones into the CXCL13-rich B-cell follicles [100,101]. In the follicles, Tfh cells provide a large panel of survival signals to B cells such as CD40L, IL-4, IL-21, programmed death-ligand 1 (PD-1), and BAFF [102] and contribute to the generation of most memory B cells and plasma cells. In SS pathology, a significantly increased number of Tfh cells were observed in the anti-SSA/SSB positive patients, as well as in patients with increased serum IL-12, IL-21 levels, and high focus score values suggesting that increased Tfh cells may play an important role in disease development and progression [103,104]. In conclusion, acting as a regulator of T cell-dependent B cell hyperactivity, the Tfh cells can prove to be a new therapeutic target in pSS disease [105].

Another particular type of T cells sharing similar characteristics as Tfh cells are CCR9^+^ Th cells [106]. The latter, similarly, to Tfh cells, express Bcl-6, IL-21, and ICOS but have the specific hallmark of displaying CCR9 and a lessened expression of CXCR5 [107]. CCR9^+^ T cells have a propensity of producing significantly higher levels of inflammatory cytokines such as IL-7, IL-21, IFN-γ, and IL-17. Both CCR9^+^ and CXCR5^+^ T cells correlate with increased B cell activity [108].

#### 3.1.4. T Regulatory Cells

The self-tolerance maintenance is regulated by several processes such as the deletion of self-reactive T cells. T-regulatory cells (Tregs) play a central role in this mechanism by regulating the immune homeostasis and suppressing autoreactive lymphocytes through cell-cell contact or the release of cytokines including IL-10 and TGF-β [109,110]. The role of Tregs is explained by emerging evidences in which the suppression of immune responses might be necessary, such as in autoimmune diseases [111,112,113]. Treg cells were firstly identified as cells expressing the alpha chain of the IL-2 receptor (IL-2Rα, CD25) on their cell surface and the ability to prevent autoimmunity in an experimental mice model [114]. The differentiation of naïve T lymphocytes into Tregs depends on the specific cytokine microenvironment and the expression of the forkhead box protein transcription factor P3 (FoxP3). In general, the presence of TGF-β and the absence of IL-6 promotes the Treg phenotype instead of Th17. Thus, TGF-β is required in both cases, but the presence or absence of IL-6 leads to the generation of Th17 or Treg cells, respectively [115]. It seems clear that the perturbation of this process could induce the generation of pathogenic Th17 cells and the development of autoimmunity. To date, the role of Treg cells in SS still remains controversial [116,117,118,119,120,121,122,123]. The observed discrepancies may, in part, have been due to the strategies used to analyze Treg cells. In fact, in the past, the proportion of circulating Treg cells was based on the expression of CD25 without taking into account the co-expression of FoxP3. Some studies have shown a reduction in CD25^high^Treg cells [117,118,120,123], but the association with clinical or serological manifestations was observed only in a few studies [117,118]. Some other studies have observed a reduction of Treg cells in peripheral blood in patients with a milder clinical picture without extra-glandular manifestations [120]. In contrast, other studies have reported an increase in circulating Treg cells in SS [121] while still others have described that the Treg proportions were comparable between pSS and controls [116,122]. Despite these conflicting evidences, a specific subset of CD4+ T cells expressing Foxp3, TGF-β, and IL-10 but low levels of CD25 have been observed in SS patients. This new subsets of CD4+ T cell is increased in patients with inactive disease compared to the control and present a strong inhibitory activity against autoreactive cells [123,124].

#### 3.1.5. Follicular Regulatory Cells (Tfr)

Follicular regulatory T cells (Tfr) are a subtype of Treg specialized in the regulation and suppression of Tfh and B cells activity [125]. Tfr express high levels of CXCR5, inducible the T cell co-stimulator (ICOS) and PD-1 on the cell surface, which directs them to follicles and GC regions [126]. In addition, Tfr cells express specific Treg markers such Bcl6 [127], Foxp3, Nuclear Factor of activated T cells 2 (NFAT-2), Glucocorticoid-induced TNFR-related protein (GITR), B lymphocyte-induced maturation protein-1 (Blimp-1), and cytotoxic T-lymphocyte-associated protein 4 (CTLA-4). The Tfr differentiation program is triggered by dendritic cells [128] and B cells [129] thanks to costimulatory signals such as CD28 and ICOS [125,130]. NFAT-2 facilitates CXCR5 up-regulation in Foxp3+ T cells [131]. Following their differentiation, Tfr cells enter the circulation to become memory Tfr or to migrate to the B cell zone [132]. In an experimental Bcl6fl/flFoxp3Cre (KO) mice model, the deletion of Tfr cells makes them susceptible to the development of autoimmune diseases and experimental Sjögren’s syndrome [133]. Periphery blood and SGs analysis from SS patients shows an enrichment of Tfr cells with a Tfr/Tfh ratio increased if compared to the control [134,135]. However, the involvement of Tfr cells in exacerbation of SS still remains controversial.

### 3.2. CD8+ T Cells

The histological analysis of SGs from pSS patients show that CD4+ T cells are the most common cell type. However, recent studies have also observed the presence of activated CD8+T cells that express a high level of human leukocyte antigen (HLA)-DR and positively correlate with several disease features [136]. In addition, the expression of C-X-C motif chemokine receptor type 3 (CXCR3) by CD8+ T cells in SS patients may be involved in the migration of them to the inflamed SGs [137]. In non-obese diabetic (NOD) mouse model of Sjögren’s syndrome, CD8+ T cells are activated and produce inflammatory cytokines. In addition, the transfer of CD8+ T cells from NOD mice lymph nodes (LNs) into NOD-severe combined immunodeficiency hosts induces the inflammation of the lacrimal glands. These results demonstrate the pathogenic role of CD8+ T to induce exocrine gland autoimmunity in NOD mice [138].

More recently, it has been suggested that CD4−CD8− (Double-Negative) T cells also play a role in the pathogenesis of pSS. These cells are expanded in several inflammatory situations and invade the inflamed tissues contributing to the tissue damage in autoimmune disease such as psoriasis, SLE, and SS [139].

#### Innate T Cells

Different innate T cells are present in the peripheral blood and/or the SGs from pSS patients. They include mainly gd, invariant NK T cells, and mucosal-associated invariant T cells. These cells are activated rapidly upon stimulation and are not dependent on MHC-II activation [140].

### 3.3. B Cells

#### 3.3.1. B Cell Hyperactivity

B-cell hyperactivity represents one of the hallmarks of the SS and also plays a key role in the autoimmunity and lympho-proliferative processes [141,142,143]. B cells originate from the bone marrow from hematopoietic stem cells. During development, B cells go through different stages of selection which exclude a substantial fraction of self-reactive and polyreactive B cell [144]. In the first checkpoint, which takes place in the bone marrow, polyreactive B cells are removed during a process known as central tolerance. In the second checkpoint, in the periphery, only a small amount of self-reactive and polyreactive mature naïve B cells could survive. The third tolerance checkpoint, known as the pre-germinal center checkpoint, allows the exclusion of self-reactive naïve B cells from entering B cell follicles [145]. A recent study has shown a deficiency in both early and late B cell tolerance checkpoints in patients with SS. In addition, B cell depletion using antibodies against CD20 in the inhibitor of DNA binding 3 (Id3) KO mice model leads to a significant histological recovery associated with an improvement of saliva secretory functions. This observation corroborates the hypothesis that B cells could play a critical role in SS exacerbation [146].

Some studies have posited an early activation of B cells in SS by the innate immune system. The interferon signaling pathway, which is a key feature in SS, interacts with B cells to trigger the production of autoantibodies [147]. A new subtype of neutrophils found in the splenic marginal zone (MZ), favored the activation and proliferation of B cells by inducing the secretion of BAFF, (a proliferation-inducing ligand) APRIL, and IL-21 [22,148].

#### 3.3.2. B Cell Subpopulations

Different B cell populations have been found in SGs as well as in peripheral blood from pSS patients [149]. There is a significant increase of IgD^+^CD38^+^ B cells expressing CD19 in pSS patients. Moreover, there is also a substantial increase in Bruton tyrosine kinase (BTK) in the B cells of pSS patients [150]. The resulting effect of these important numbers of B cells is enhanced B cell signaling through B cell receptors. Besides the role of memory B cells, plasmablasts and plasma cells have been implicated in the pathogenesis of pSS. In pSS patients with lymphoma, increased numbers of CD27 ^high^CD19 ^low^CD20^−^ plasmablasts have been identified [151]. The immunophenotyping study of pSS defined six different subcellular types including increased plasmablasts, activated CD4+ and CD8+ T cells, as well as decreased numbers of CD27^+^ B cells, CD4+ T cells, and pDC [152]. In the SG of pSS, immunophenotyping revealed B cells, CD8+ T cells, and activated epithelial cells. Moreover, the increased number of plasmablasts and plasma cells strongly correlated with disease activity as well as serum IgG and the presence of auto antibodies [152]. In SG from pSS patients, there was an important proportion of infiltrating B cells that were fully differentiated plasma cells [152].

#### 3.3.3. Marginal Zone B Cells

Marginal zone B cells are a subset of splenic B cells. Their involvement in the pathophysiology of pSS has not yet been demonstrated. An increased number of marginal zone-like B cells in the MSG of pSS patients has been observed [153]. In addition, studies using mice models of pSS showed that the depletion of MZ B cells prevented the development of pSS manifestations in mice [154,155,156,157]. Even if there is no current evidence of the importance of MZ B cells in pSS patients, an increased percentage of CD27^+^ memory B cells from the SGs of the latter were identified as IgM^+^ cells [153]. Furthermore, the fact that pSS associated lymphoma stem from mucosa-associated lymphoid tissue (MALT) tissue, fuels the role of this subtype of B cells in pSS [158].

#### 3.3.4. Regulatory B Cells

Regulatory B cells are another type of B cell involved in pSS. These types of B cells exert their regulatory function through the production of IL-10. As such, B regulatory cells by inducing the expansion of Treg cells and by the actions of IL-10, are able to control the proliferation of Th1 cells and restrain the actions of TNf-α, Th17 cells IL-12 producing dendritic cells as well as CD8+ T cells. Beyond the production of IL-10, it has been recently shown that some B cells can also secrete IL-35. IL-35 is a cytokine which is part of the IL-12 family and has anti-inflammatory and immunosuppressive properties. In pSS, there is a disequilibrium between IL-12 and IL-35 in favor of IL-12. There was a significant decrease in serum IL-35 in pSS patients that was inversely correlated with disease activity. It has been hypothesized that the IL-12/IL-35 axis could play a noteworthy role in disease perpetuation in pSS [159].

#### 3.3.5. BAFF

BAFF is a pivotal cytokine that promotes the maturation, proliferation, and survival of B cells. In pSS, BAFF plays an important role in favoring the activation and proliferation of B cells, thereby leading to the production of autoantibodies. In essence, BAFF is secreted by dendritic cells, monocytes, and macrophages but in pSS both T and B cells as well as epithelial cells have been shown to release BAFF.

In pSS patients, activated SGECs are able to produce BAFF, undergirding their role in the pathogenesis of the disease as well as demonstrating the link between innate and adaptive immunity. This is illustrated and strengthened by studies showing elevated levels of BAFF in the sera of pSS patients as well as its correlation with anti-Ro/SSA and anti/La-SSB and rheumatoid factor (RF) [160]. Furthermore, a strong correlation between BAFF secretion by monocytes and type I IFN was found. A role for BAFF in the formation of ectopic germinal centers (GC) has also been advocated [161] but nullified by further studies implying that other pathogenic pathways might be involved [162].

#### 3.3.6. Germinal Center-Like Structures

The presence of B cells in SGs is an important characteristic of pSS extending from a discrete infiltrate to the formation of the ectopic germinal center (GC) completely invading the glands. Ectopic GC-like structures play a major role in the pathogenesis of pSS by favoring chronic B cell activation. They are present in 10–30% of pSS patients, are associated with the presence of autoantibodies anti-SSA, as well as an increased risk of developing lymphoma [163]. CXCR5 and CXCL13 are mainly involved in the formation of GC-like structures by recruiting Tfh and B cells [164].

## 4. Cross Talk between the Innate and Adaptive Immunity

The pathophysiology underlying pSS is complex with different intricating players from the innate immunity cells and the adaptive immunity T and B cells. Figure 3 illustrates the orchestrating of these different tenants interacting together to trigger the initiation and perpetuation of disease.

In genetically susceptible individuals, together with the presence of a hormonal disequilibrium as well as environmental factors such as viruses, there is an activation of the epithelial cells. Following the activation of epithelial cells, there is an upregulation of innate immune cells such as TLRs and ensuing pro-inflammatory cytokine production. Furthermore, activated and injured SGECs release exosomes and apoptotic blebs containing ribonucleoprotein autoantigens such as Ro/SSA and La/SSB that can attract both classical DC and pDC within SGs.

The activation of DC and pDC can trigger the production of type 1 and type 2 Interferons. The production of IFN-α by pDC and the secretion of inflammatory cytokines by conventional DC such as IL-12 and IFN-γ can induce tissue damage. The production of IFN-α as well as IFN-γ can enhance the secretion of BAFF thereby fostering B cell activation and proliferation as well as the secretion of auto antibodies. The production of autoantibodies can form immune complexes with the autoantigens (Ro/SSA and La/SSB) and can further thrust the secretion of IFN-α, thereby constituting a vicious inflammatory loop perpetuating disease progression.

It has to be stressed that the time of the differentiation factor expressions, different cell activations, as well as chemokine signaling and homing of lymphocytes is, perhaps, the most critical for the development, perpetuation, and severity of the clinical phenotype.

## 5. Conclusions

The innate and adaptive immunity are central in the pathogenesis of pSS, each of them representing a multi-step process leading to the triggering and perpetuation of disease. The salivary epithelial cells are at the heart of disease process, featuring as the main initial triggers of the disease and maintaining the crosstalk with B and T cells, thereby further relating to the production inflammatory cytokines and the production of autoantibodies. Further knowledge on deciphering and dissecting the different intricate players of the pathogenesis of pSS could pave the way for new avenues of therapies.

## Figures and Tables

**Figure 1 ijms-22-00658-f001:**
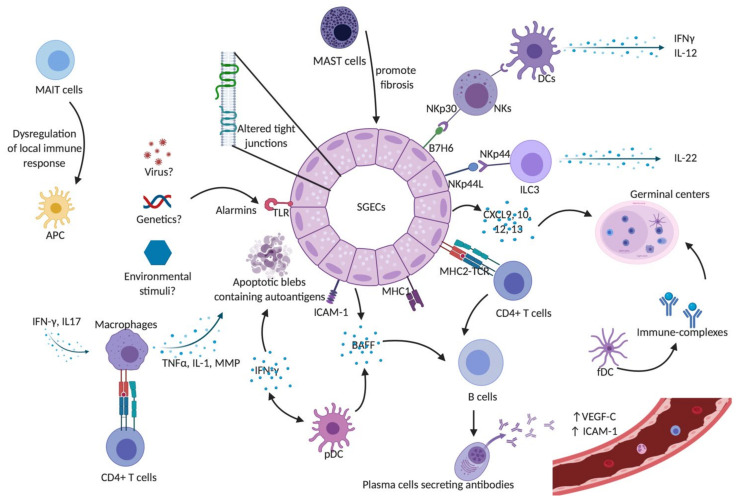
Innate immunity in Sjögren’s syndrome. SGECs constitute the main target of auto-immunity in pSS, described as an autoimmune epithelitis. SGECs exhibit a subverted architecture mainly characterized by altered tight junctions. In genetic susceptible subjects, environmental stimuli such as viruses may trigger salivary gland epithelial cells (SGECs) through TLR activation. Activated SGECs secrete the BAFF that promotes activation and maturation of B cells. SGECs also produce chemokines such as CXCR9, 10, 11, and 12 that attract immune cells and contribute to the formation of germinal centers. Activated SGECs have the ability to act as non-professional antigen-presenting cells by expressing MHC-I, (HLA-ABC) and MHC-II (HLA-DR), adhesion molecules such as ICAM1 allowing them to activate T cells. TLR activation also contributes to SGEC apoptosis, releasing autoantigens that drive autoimmunity in pSS. Activated macrophages produce inflammatory cytokines such as IL-1, TNFα, and MMPs leading to epithelial cell damage. They can also act as antigenic peptide presenting cells through their MHC-II and interact with antigen-specific CD4+ T cells. pDCs lead to the production of type I IFN that acts through autocrine and paracrine circuits feeding a continuous reinforcing inflammatory loop. It also induces the production of BAFF, production contributing to the activation of B cells into plasma cells. fDCs play an essential part in the structure of ectopic germinal centers and retain on their surface immune-complexes, formed by antigen-antibody-complement. Mast cells contribute to the fibrosis and fatty infiltration of salivary glands (SGs). The aberrant phenotype of MAIT cells in pSS patients may lead to the dysregulation of the local immune responses, which would trigger local damage in SGs and auto-immunity. NK cells express the NKp30 receptor that is recognized by DCs and lead to the production of Th1 cytokines such as IFN-γ and IL-12. SGECs express B7-H6, the ligand for NKp30. Taken together, this may explain the hyperactivity of NK cells and the cross-talk with SGECs and DCs that lead to a subsequent activation of innate and adaptive immunity. A subset of ILC3 was found to be a major source of IL-22 in SGECs. Abbreviations: APC: Antigen presenting cells; BAFF: B-cell activating factor; CXCL9: C-X-C motif chemokine type 9; CXCL10: C-X-C motif chemokine type 10; CXCL12: C-X-C motif chemokine type 12; CXCL13: C-X-C motif chemokine type 13; DCs: dendritic cells; fDCs: follicular dendritic cells; ICAM-1: intercellular adhesion molecule 1; IFN-γ: interferon gamma; IL-: interleukin; ILC3: innate Lymphoid Cells type 3 MAIT: Mucosal-associated invariant T cells; MHC-I: major histocompatibility complex class I; MHC-II: major histocompatibility complex class II; MMPs: metalloproteases; NK: natural killer cells; NKp44L: NKp44 ligand; pDCs: plasmacytoid dendritic cells; SGECs: salivary glands epithelial cells; TCR: T cell receptor; TLR: Toll like receptor; TNFα: tumor necrosis factor alpha; VEGF-C: vascular endothelial growth factor C.

**Figure 2 ijms-22-00658-f002:**
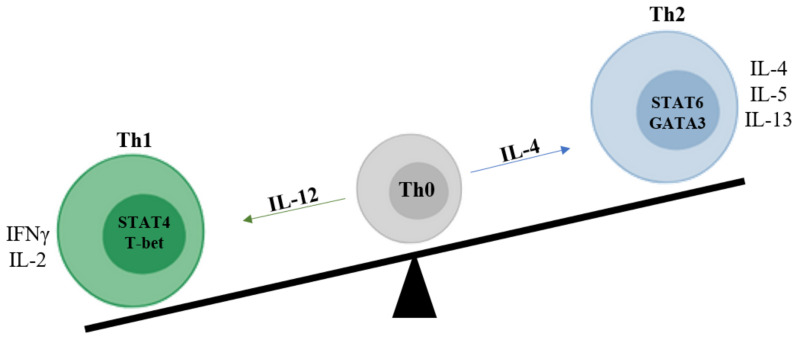
Th1-Th2 imbalance. Upon T-cell activation, IFN-γ, and IL-12 induce the expression of T-bet and STAT-4, which is involved in the differentiation of naïve CD4+ T cells into Th1 lymphocytes. Th1 cells predominantly produce pro-inflammatory cytokines such as IFN-γ and IL-2. In contrast, IL-4 induces the GATA-3 transcription factor and the consequent polarization of naïve T cells into Th2. Th2 cells produce anti-inflammatory cytokines such as IL-4, IL-5, and IL-13. Several studies have suggested that pSS is related to abnormal Th1 activation and SGs infiltration. It is supported by the presence of elevated levels of IFN-γ in serum and Th1 cells in blood. Furthermore, T cells expressing a high level of IFN-γ and STAT-4 mRNA have been found in SGs from pSS patients. This Th1/Th2 imbalance, generally observed in various chronic inflammatory disorders, is not easily understood because of a limited number of studies. Abbreviations: IFN-γ:interferon gamma; IL-: interleukin; pSS: primary Sjögren’s syndrome; STAT: signal transducer and activator of transcription; T-bet: T-Box Transcription Factor 21; Th1: type 1 helper cells; Th2: type 2 helper cells.

**Figure 3 ijms-22-00658-f003:**
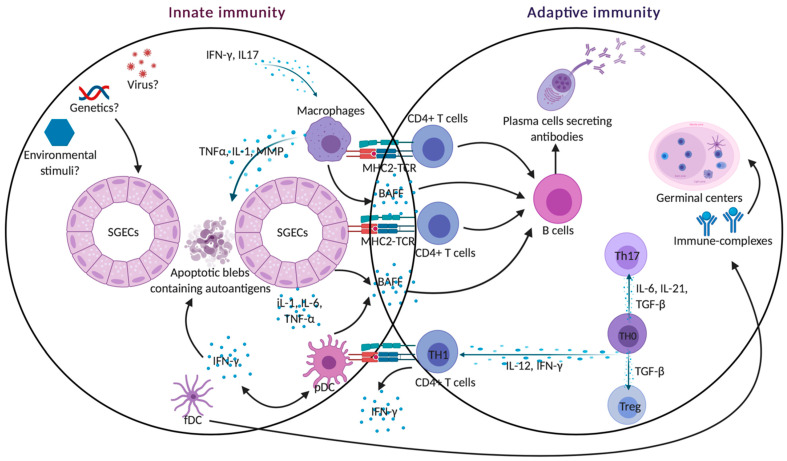
Innate and adaptive crosstalk. Activated SGECs secrete BAFF that promotes the activation and maturation of B cells into long-lasting memory B cells and plasma B cells producing auto-antibodies. SGECs also produce chemokines IL-1, IL-6, IL18, and TNFα that attract immune cells and contribute to the formation of germinal centers. Activated SGECs have the ability to act as non-professional antigen-presenting cells by expressing MHC-I (HLA-ABC) and MHC-II (HLA-DR) adhesion molecules such as ICAM1, allowing them to activate T cells. TLR activation also contributes to SGEC apoptosis, releasing autoantigens that drive autoimmunity in pSS. Activated macrophages can act as antigenic peptide presenting cells through their MHC-II and interact with antigen-specific CD4+ T cells. pDCs lead to the production of type I IFN that acts through autocrine and paracrine circuits feeding a continuous reinforcing inflammatory loop. It also induces BAFF production, contributing to the activation of B cells into plasma cells. DCs also play an essential part in the structure of ectopic germinal centers and retain on their surface immune-complexes, formed by antigen-antibody-complement.

## Data Availability

Not applicable.

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
