# Peer review of "The Involvement of Innate and Adaptive Immunity in the Initiation and Perpetuation of Sjögren’s Syndrome"

_ijms, 2021, doi:10.3390/ijms22020658_

Round 1

Reviewer 1 Report

In this manuscript, Dr. Chivasso and colleagues review various innate and adaptive immune processes, primarily at the cellular level, considered important elements in the pathophysiological attributes underlying Sjögren´s Syndrome (SS).  SS, when compared to many other autoimmune pathologies, is a highly complex disease, as indicated by the historical difficulty in forming a clear concensus in definition.  Not surprising therefore that nearly 4,000 review articles have been published pointing to various potential underlying pathophysiological processes and/or interactions considered important to disease development, severity and heterogeneity.

In the current manuscript by Dr. Chivasso et al., the authors have taken a somewhat global approach that ties the innate and adaptive responses together to highlight the importance of each in order to obtain a more comprehensive and unified picture.  The approach of singling out individual immunity-associated cell populations and cellular processes controlled by these individual cell populations, followed by potential interactions, provides the reader with how these individual processes may influence initiation and development of the overall picture.  Thus, despite complexity in the overall subject, in the end, the reader can easily visualize the overall and stepwise development of the autoimmune response as proposed by the authors. From this point of view, the manuscript is well-written and informative.

While one can appreciate the simplification of the bioprocesses discussed, there are a couple of áreas that appear to be over-simplified.  For example, while NK and iLC cell populations have been identified within the infiltrates present in SS patient´s salivary glands, considering the relatively small numbers, one must ask if they are truly critical/necessary to disease onset (?) despite the published reports.  Would one see SS develop if neither cell were present?  Seeing cells of any immune population in an inflammatory lesión may not be an unusual situation, e.g., multiple mouse lines develop lymphocytic infiltrates in their salivary glands without a clinical disease.  Similarly, suggesting that SS may be related to an inbalance of Th1 : Th2 (Fig.2) and invoking differences in IFN/IL12 vs IL4/IL13 as the reason for SS development is an interesting  (overstated) message since SS requires the production of IL4, most likely produced by CD4+ T cells, to actívate B cell populations that in turn are also required for SS development.  Thus, it seems important to point out that the time of differentiation factor expressions, various cell activations, and chemokine signalings are most critical for development, persistance and severity of clinical disease.  In essence, is it not more critical for the timing of these immune cell activations.  Lastly, neither Fig.2, Fig.3 nor the Text provide the reader with information on the critical issue of B cells acting as antigen-presenting cells in both the marginal and folicular zones of the spleen (see Roark et al., J Immunol 1998, 160:3121 and Mariño et al., Diabetes 2008, 57:395), not just antibody production. Antibody production is only one important role for B cells in SS.

Overall, this Review is an excellent and broad discussion of the topic.  It presents a nice overview that relates the innate and adaptive responses present in the autoimmune response underlying SS as interactive partners.  A few over-simplifications, especially with respect to the nature and role of B cells, should be addressed.     

Author Response

Dear Editor,

We heartily thank the reviewers for their useful and witful comments for improving our manuscript entitled: The involvement of innate and adaptive immunity in the initiation and perpetuation of Sjögren’s syndrome.

Please find below our timely and appropriate responses.  All the comments and suggestions were implemented in the main manuscript.

For the response to Reviewer 1:

  • Regarding Natural Killer cells and Innate Lymphoid Cells, we have further discussed the lack of data concerning their true role in the initiation of the disease.
  • We have highlighted and added the importance of timing in the development, persistence and severity of clinical disease regarding the various cells involved.
  • We have further discussed and extended the paragraph with respect to the nature and role of B cells. By itself, the role and involvement of B cells are pivotal in the development of Sjögren’s disease and therefore could constitute a full review. We have delved in the matter of B cells in disease pathology but have tried to be as clear as possible.

    Kind regards

    On Behalf of all the co authors

Reviewer 2 Report

Chivasso et al’s review “the involvement of innate and adaptive immunity in the initiation and perpetuation of Sjogren’s syndrome” tackles a broad review of individual innate and adaptive cell types that contribute to the pathogenesis of SS. Each cell is generally defined and then fit into the pathogenesis of SS with broad descriptive figures to accompany the text. Previous studies have similarly provided a cell-based break down of SS pathogenesis (pmid 28330683); however, this review includes a broader range of cell types in both innate and adaptive immune systems, slightly expanding the knowledge presented in previous review publications. There are many grammar errors that will require correction.

  • Major
    1. Sjogren’s syndrome
      1. Remove or/and anti-La/SSB. Based on studies by Baer et al. in ann rheum disease, SSB no longer considered as relevant
    2. 2 Macrophages
      1. Increase discussion on macrophages on SS. For example, recommend inclusion of data on macrophages and correlation with lymphoma (i.e. Kapsogeorgou ek., et al. J. Rheumatol. 2013;40)
    3. 3. Mast cells
      1. Include more recent publications on SS and fibrosis of SG (leehan et al. clin exp rheumatol 2018).
      2. Mention IL33 as well
    4. 4 SGEC
      1. Discuss apoptotic bleb mechanisms here as exposure mechanism to autoantigens. This mechanism is mentioned later in the section 5 crosstalk but should also be mentioned here in detail.
      2. Mention how tlr stim also regulates SSA within SGEC (Kyriakidis N. C. et alClinical and Experimental Immunology. 2014)
      3. Define what “exogenous aggression” in context of immune activation
  1. Adaptive immunity
    1. Clarify that B cells are also involved early in more severe disease. Christodoulou et al disease. J Autoimmun 2010;34:400-7
    2. T cells are important but perhaps not “the central protagonist” as posited. SGECs are also proclaimed as cardinal. Reduce strength of this statement
  2. 1.8 B Cells hyperactivity
    1. Include subheadings akin to T cell subsections
    2. Re-phrase “b-cell hyperactivity represents one of the most frequent hallmarks…” this phrase competes with previous comments about hallmark and cardinal features of pathogenesis.
    3. Similarly, overuse cardinal (i.e. principal) for multiple pathways including IL-12/IL-35, SGECs, and BAFF.
    4. Cite data for “a role for baff in ectopic GC has also been advocated…”
  • Minor
    1. abstract
      1. instead of “mainly” use “including” salivary and lachrymal glands
      2. …interstitial lung involvement, AND neurological involvement
    2. Sjogren’s syndrome
      1. Remove term “prevalently” and instead “including”
      2. Include commas: ….primary SS (pSS), which occurs in the absence of other autoimmune diseases, and secondary SS (sSS), which is associated with other autoimmune.
      3. Sex instead of sexual
      4. Use male and female not man and women (male and female denotes sex chromosome derivation but women and men have self-defined connotations)
  1. 3 Mast cell
    1. Fix sentence “mast cells are immune cells almost constutively found in connective tissues”
    2. Mixed past and present tense. Keep present tense as this is more consistent with remainder of documentFix “mast cells activation through tlr-2…”
    3. Remove semi colon and replace with comma. Which is always preceded by comma.
  1. 4 SGEC
    1. Do not bold “furthermore”
  2. 5 endothelial cells
    1. “Were reported from minor salivary glands (msgs) from pSS patients
    2. “…aberrant lymphatic neovascularization leads to a persistent extravasation of immune cells.
  3. 6 Mait cells
    1. Wang et al found that MAIT cells are decreased in peripheral blood circulation but are increased in SGs…
  4. 7 NK cells
    1. NK cells are a cytolytic component of the innate immune system
  5. 3 adaptive immunity
    1. Remove phrase “debulked” and replace with different term
    2. Write CD4+ T lymphocytes not T lymphocytes, CD4+. Please do the same with the following CD8 T lymphocytes expression.
  6. 1.1 th1-th2 cells
    1. remove “for a long time”
    2. Remove remaining comment mark
  7. 1.2 TH17
    1. “55% amino acid sequence identity”
    2. Rearrange citations in sentence-citation 86 seems out of place
  8. 1.3 T follicular helper cells
    1. Adjust phrase so easier to read “Tfh cell differentiation canonical process requires…”
    2. The best marker that defines…
    3. …high focus score values that suggest increased Tfh…
  1. 1.7 innate t cells
    1. Remove remaining comment
  2. 1.8 B cell hyperactivity
    1. Remove word “sheer”
  3. Cross talk
    1. Replace entail with another word
  4. Other
    1. Numbering goes from 3.1.8 to 5. This should be changed to sequential

Author Response

Dear Editor,

We heartily thank the reviewers for their useful and witful comments for improving our manuscript entitled: The involvement of innate and adaptive immunity in the initiation and perpetuation of Sjögren’s syndrome.

Please find below our timely and appropriate responses.  All the comments and suggestions were implemented in the main manuscript

Please find below the responses to reviewer 2:

Major

1.Sjogren’s syndrome

  • Or/and anti-La/SSB has been removed from the text
  • Macrophages
    • Discussion on macrophages has been implemented. We have included data on macrophages and correlation with risk of lymphoma development.
  • Mast cells
    • More recent publications on SS and fibrosis of SG have been included, as suggested Leehan et al. Clin exp rheumatol 2018.
    • IL33 has been discussed as well and implemented.
  • SGEC
    • The mechanisms underlying apoptotic blebs have been added.
    • The role of TLR stimulation regulating  SSA within SGEC has been discussed 
    • Exogenous aggression in context of immune activation has been detailed.
  1. Adaptive immunity
  • The role of B cells and its relation to innate immunity to early and severe disease phenotype has been discussed.
  • T cells are important but perhaps not “the central protagonist” as posited. We softened the weight of this statement.

3.3. B Cells

  • Subheadings have been added
  • The phrase “b-cell hyperactivity represents one of the most frequent hallmarks…” has been modified to avoid competition with previous comments about hallmarks and cardinal features of pathogenesis.
  • We fixed the overuse of the word “cardinal”
  • We have cited data for “a role for Baff in ectopic GC has also been advocated…”

Minor:

All the minor comments and suggestions as well as grammatical mistakes have been adapted in the text accordingly.

We thank the reviewers for their comments and suggestions that have contributed to improving the quality of the manuscript.

Kind regards

On Behalf of all the co authors

Round 2

Reviewer 2 Report

No further comments